# Determinants of Working Practice Location for Clinicians According to High School, Medical School, and Resident Training Locations in Korea

**DOI:** 10.3390/healthcare11091203

**Published:** 2023-04-22

**Authors:** Kyungah Park, Hyeongsu Kim, Jeehye Lee, Jinyoung Shin, AhHyun Park

**Affiliations:** 1Department of Preventive Medicine, Konkuk University School of Medicine, Seoul 05029, Republic of Korea; 2National Emergency Medical Center, National Medical Center, Seoul 04564, Republic of Korea; 3Department of Family Medicine, Konkuk University School of Medicine, Seoul 05029, Republic of Korea; 4Expert Group on Health Promotion for Seoul Metropolitan Government, Konkuk University, Seoul 05029, Republic of Korea

**Keywords:** retention rate, consecutive medical training, geographical maldistribution of physicians, Korean physician survey

## Abstract

Although several regulations have been implemented for medical school admission, such as a quota system, the uneven distribution of healthcare personnel across regions is an unresolved problem in Korea. This study explores the distribution and retention rate of clinicians across regions according to the degree of experience staying in the current clinical area during high school/medical school/resident training using 2016 Korean Physician Survey data. Both in metropolitan and non-metropolitan areas, clinicians who completed high school, medical school, and resident training in the current practice region (Subgroup D) accounted for the largest proportion (Metro, *n* = 1611, 46.1%; non-metro, *n* = 1917, 52.9%). The retention rate was the highest in Subgroup D both in metropolitan (84.3%) and non-metropolitan areas (Chungcheong 86.2%, Jeolla 79.9%, Daegu/Gyeongbuk 81.6%, Busan/Ulsan/Gyeongnam 93.3%) except Gangwon and Jeju. The second, third, and fourth highest retention rates were observed in cases where clinicians completed their high school and resident training, medical school and resident training, and resident training only, respectively, in all regions, although the ranking differs by region. To increase the retention rate of physicians, this study shows that it is necessary for a student to seek ways to continue training in the same region in which they graduated from medical school.

## 1. Introduction

The geographical maldistribution of physicians is a significant global problem hindering equitable access to healthcare services [1]. There is typically an imbalance between the urban and rural areas in many countries, and various policies are being implemented to address this issue [1].

In Korea, 22 of 41 medical schools were established after 1980. Among these, 17 were established in non-metropolitan areas, while the remaining 5 were established in metropolitan areas of Gyeonggi and Incheon with the aim of resolving the imbalance of medical facilities and resource distribution between regions. Despite these efforts, the uneven distribution of healthcare personnel across regions remains unresolved.

In addition, there is a very high demand for admission to medical school in Korea, and about half of all medical students enrolled in non-metropolitan schools are, in fact, from metropolitan areas [2]. On the other hand, clinicians prefer to work in cities [3]. Moreover, as Korea is relatively small, and it is easy to move between regions, differences in the retention rate of clinicians by region are expected, particularly between metropolitan and non-metropolitan areas.

Several studies have investigated the factors influencing physicians’ choice of practice location. Physicians with a rural background, whether in terms of their hometown, medical school, or resident training, are more likely to practice in rural areas [4,5,6,7,8,9]. However, in Korea, there is a lack of studies regarding the potential impact of consecutive regional exposure (i.e., the experience of having grown up, studied, and completed resident training in the same region) on the selection of practice location. This study explores this topic.

## 2. Materials and Methods

### 2.1. Study Design and Data Source

Secondary data analysis was conducted using 2016 Korean Physician Survey (KPS) data to determine whether physicians who have remained in the same region as their hometown through their medical school and resident training are more likely to settle in that area. The 2016 KPS was the first national survey of all medical doctors in Korea, and it was conducted as a web-based self-administered questionnaire survey by the Research Institute for Healthcare Policy (RIHP) of the Korean Medical Association (KMA). This was a complete enumeration survey targeting doctors who agreed to disclose personal information among all doctors registered in the KMA. In order to secure representativeness, the database of the members of the Korean Medical Association was used as a sampling frame, and the target population was formed and surveyed in parallel with the Survey Stratified Quota sampling method, which uses the distribution by gender, age, and occupation as stratification variables. The survey was conducted from 21 November 2016 to 8 January 2017, during which period, emails were sent to 77,997 of 108,870 medical doctors who registered their information in the KMA database, excluding 30,873 that did not agree to disclose their personal information or email addresses. In all, 8564 members (13.8%) participated in the 2016 KPS [10].

### 2.2. Selection of Study Population

Of the 8564 respondents, we excluded 922 who answered that their current employment status was nonclinician, retiree, intern, or resident to analyze the distribution of incumbent clinicians. We also excluded those who graduated from high school outside of Korea (*n* = 38), who had insufficient high school information (*n* = 107), who graduated from medical graduate schools (*n* = 191), who graduated from medical schools outside of Korea (*n* = 13), who were working outside of Korea (*n* = 11), who did not have available workplace information (*n* = 65), who were not qualified as specialists (*n* = 446), and who were under 30 years old (*n* = 262). Therefore, the final study population consisted of 7122 participants (Figure 1).

### 2.3. Variables

The dependent variables were the following subgroups (Figure 2), including A (those who completed high school and medical school education in the current practice region (CPR)), B (completed medical school and resident training in the CPR), C (completed high school and resident training in the CPR), D (completed high school, medical school, and resident training in the CPR), E (completed only high school in the CPR), F (completed only medical school in the CPR), G (completed only resident training in the CPR), H (did not complete any education course in the CPR).

The independent variables were sex (male/female), age (30–39, 40–49, 50–59, and ≥60 years), marital status (never married, married, or unmarried), current employment status (owner of clinic or hospital, paid doctor, or medical professor), and specialty (medical, surgical, or support). The medical specialties included internal medicine, neurology, psychiatry, pediatrics, dermatology, tubercular medicine, rehabilitation, and family medicine. The surgical specialties included general, orthopedic, thoracic, plastic, obstetric and gynecological, ophthalmological, otorhinolaryngological, urological, neurosurgery, and emergency medicine. Support medicine consisted of anesthesiology and pain management, radiology, radiation oncology, pathology, laboratory medicine, preventive medicine, nuclear medicine, and occupational and environmental medicine.

### 2.4. Data Analysis

The data were analyzed using SAS version 9.2 (SAS Institute Inc., Cary, NC, USA).

To classify regions, 17 administrative divisions of South Korea were reclassified into 7 regions, including the metropolitan region (Seoul, Gyeonggi, Incheon, Republic of Korea), Chungcheong region (Daejeon, Sejong City, Chungcheongnam-do, Chungcheongbuk-do, Republic of Korea), Jeolla region (Gwangju, Jeollanam-do, Jeollabuk-do, Republic of Korea), Gangwon region (Gangwon-do, Republic of Korea), Daegu/Gyeongbuk region (Daegu, Gyeongsangbuk-do, Republic of Korea), Busan/Ulsan/Gyeongnam region (Busan, Ulsan, Gyeongsangnam-do, Republic of Korea), and Jeju region (Jeju-do) (Appendix A). Six regions outside of the metropolitan region were classified as non-metropolitan areas.

Next, the proportion of each subgroup of the 7 regions was calculated, and frequency analysis was performed to determine the relations between dependent and independent variables using the Mantel–Haenszel chi-square test with *p* < 0.05 taken to indicate significance.

Finally, we calculated the retention rates to determine how many clinicians remained in the regions where they finished their high school/medical school/resident training education. Retention rate was defined as the percentage of clinicians who completed their education and training in a specific region and continued to practice in the same region. We divided the study population into 8 subgroups based on the region where they finished their high school/medical school/resident training among the 7 regions outlined above.
$$Retention\;rate\;(\%)=\frac{Number\;of\;clinicians\;who\;remained\;in\;the\;same\;region}{Number\;who\;finished\;high\;school,\;med\;school,\;and\;resident\;training\;in\;the\;region}\;\times \;100$$

### 2.5. Ethics Statement

The need for study protocol review was waived by the Institutional Review Board of Konkuk University due to the retrospective nature of this study. Patients and the public were not involved in the study design, data collection, analysis, or interpretation of data (7001355–202203-E-164).

## 3. Results

### 3.1. Proportions of Each Subgroup in Metropolitan and Non-Metropolitan Regions

Among all subjects (*n* = 7122), Subgroup D accounted for the largest proportion of respondents (*n* = 3528, 49.5%), followed by Subgroups H (*n* = 1062, 14.9%) and B (*n* = 723, 10.2%), with Subgroup F accounting for the smallest proportion (*n* = 75, 1.1%). In the metropolitan region, Subgroup D accounted for the largest proportion (*n* = 1611, 46.1%) followed by Subgroups G (*n* = 603, 17.2%), B (*n* = 548, 15. 7%), and C (*n* = 409, 11.7%) among the 3497 clinicians practicing. In the non-metropolitan regions, Subgroup D accounted for the largest proportion (*n* = 1917, 52.9%), followed by Subgroups H (*n* = 849, 23.4%), A (*n* = 296, 8.2%), and E (*n* = 182, 5.0%). In the case of the non-metropolitan regions, the rankings differed slightly by region, but Subgroup D, Subgroup H, and Subgroup A accounted for the top 3 proportions in all regions, except for Gangwon (*n* = 194) and Jeju (*n* = 90), where the number of subjects was relatively small (Table 1).

### 3.2. Relations between Subgroups and Sociodemographic Characteristics

The relationship between the subgroups and sociodemographic characteristics is shown in Appendix A. Sex (*p* = 0.0178), age (*p* = 0.0021), and specialty (*p* = 0.0024) were significantly related to the subgroup distribution. Marital status (*p* = 0.4769) and employment status (*p* = 0.1254) were not related to the subgroup distribution.

### 3.3. Retention Rates in Metropolitan and Non-Metropolitan Regions

Table 2 shows the retention rates of clinicians in metropolitan and non-metropolitan regions by subgroup. The retention rate in the metropolitan region was the highest at 84.3% in Subgroup D, which consisted of those who completed high school, medical school, and resident training in the metropolitan region, followed by Subgroups C (77.0%), B (71.6%), G (58.9%), and A (46.9%). In the non-metropolitan areas, Subgroup D also shows the highest retention rates in Chungcheong (86.2%), Jeolla (79.9%), Daegu/Gyeongbuk (81.6%), and Busan/Ulsan/Gyeongnam (93.3%), except Gangwon and Jeju. Except for Jeju and Gangwon, the second, third, and fourth highest retention rates in non-metropolitan regions were observed in Subgroups C, B, and G, although the ranking differs by region.

## 4. Discussion

This study explores the distribution of clinicians across regions according to the degree of experience staying in the current clinical region during high school/medical school/resident training. We also calculated the retention rate of clinicians in the region where they completed their education.

The locations of the physicians’ hometown, medical school, and resident training significantly influence their selection of a clinical practice region [4,5,6,7,8,9]. A tendency for physicians with rural experience to practice in rural areas more often than those without such experience was reported. However, no studies to date have compared retention rates according to clinicians’ degree of experience staying in a given area (e.g., having lived, studied, and trained in a non-metropolitan area).

We found that clinicians were most likely to practice in the region in which they completed all three stages of their education, including high school, medical school, and resident training. Except for Jeju and Gangwon, the second, third, and fourth highest retention rates were observed in cases where clinicians completed their high school and resident training, medical school and resident training, and resident training only, although the ranking differs by region. This is consistent with the findings of McGrail and O’Sullivan (2021), who reported that doctors with more than 1 year of rural training or 3–12 months of rural training are more likely to work in the same rural region compared to those who have less than 12 weeks of rural training [8]. These results suggest that to increase the retention rate of clinicians in a particular region, it may be beneficial to facilitate the entry of local talent (e.g., high school graduates) into medical schools in the same region, and to provide resident training opportunities in the area where they completed their medical education.

In fact, a local quota system has been implemented in Korea as a recommendation for medical school admissions since 2015. The purpose of this system is to address the quantitative imbalance of physicians between the metropolitan area and other provinces, with the hope that more local talent will choose to stay in their region after graduating from medical school. As of 2023, this recommendation will become compulsory for medical school admissions. This system currently mandates that approximately 40% of medical school freshmen spaces be filled with local students in all regions except for Jeju and Gangwon, where the percentage is 20% [11]. To evaluate its effectiveness, it is essential to track the career trajectories of medical school graduates and identify the primary reasons for their migration to other regions. Furthermore, complementary measures should be developed to promote the retention of local physicians.

Despite these efforts, some critics have raised concerns that the quota system could be viewed as reverse discrimination against metropolitan students and that it may only have a limited impact on the equitable distribution of physicians, unless other factors that facilitate the voluntary placement of physicians in non-metropolitan areas are taken into consideration.

We also found that clinicians are more likely to stay in the region where they complete their resident training compared to the region where they complete their high school or medical school education. However, current conditions in non-metropolitan areas are not conducive to retaining graduates of medical schools for resident training. Comparisons of the number of residents to the number of medical school students in each region indicated that metropolitan areas have a ratio of 1.91, whereas non-metropolitan areas have a ratio of only 0.64 (Appendix A). Consequently, despite the willingness of graduates in non-metropolitan areas to remain in their region, only about two thirds are able to stay, leaving one third with no option but to leave the area for specialist training. While the distributions of the population and specialists are similar, the residents are mainly concentrated in the metropolitan areas.

Hence, it is important to consider one’s current location when assigning a resident to a training institute in the future. This could increase the likelihood of retaining local medical graduates in each respective region.

Finally, in addition to thinking about how to increase continuous local experience in the educational process, it is necessary to think about how to voluntarily move clinicians to rural areas regardless of whether or not they have local experience. Where a clinician chooses to practice is influenced by multiple factors, including personal and financial considerations, working conditions, as well as previous educational and training experiences [12]. Therefore, it is important to investigate various policies that can encourage physicians to voluntarily establish practices in non-metropolitan regions. Examples of such policies include offering incentives for employment and settlement when local college graduates engage in clinical activities in the area, as in Canada [13] and Australia [14].

This study had some limitations in the categorization of data and analysis method, and the interpretation of the results. First, some environmental factors were not captured in the questionnaire, such as children’s educational environment or economic conditions that could affect the choice of clinical practice region. This could be overcome by conducting studies on cohorts of physicians followed up after graduation from medical school. Second, although we sampled a relatively large range of physicians, the sample sizes of some subgroups in the seven regions were insufficient to accurately determine the retention rate. This could be overcome by repeated studies or by including larger sample sizes. The strength of this study lies in the use of data from the national physician survey. Policies that aim to solve real-world problems based on actual data will be effective and serve as the foundations for more proactive problem-solving methods.

## 5. Conclusions

The lack of balanced distributions of physicians across a country is a common concern worldwide, and a number of policies and regulations have been implemented in various countries to address this issue. Korea has implemented a quota system for medical school admission, which has recently been made compulsory. However, there is controversy regarding the policy. To increase the retention rate of physicians in non-metropolitan areas, this study showed that it is necessary to seek ways to continue training in the same region in which a student graduates from medical school. This may entail increasing the resident training capacity in a given region and/or securing local quotas for medical school admission for local talent. Our results also suggest the need to explore various policies to encourage clinicians to voluntarily settle in their region of study, such as providing incentives for employment and settlement.

## Figures and Tables

**Figure 1 healthcare-11-01203-f001:**
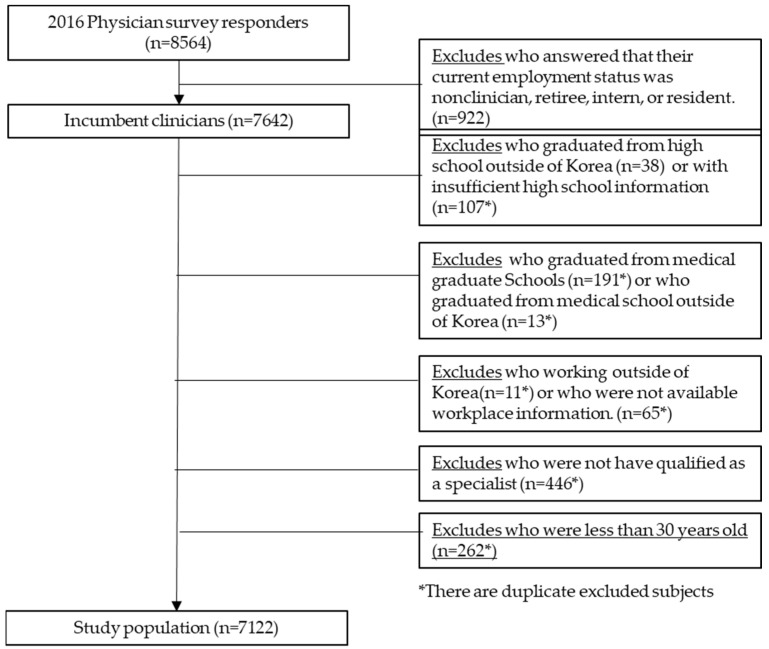
Flowchart of study population selection.

**Figure 2 healthcare-11-01203-f002:**
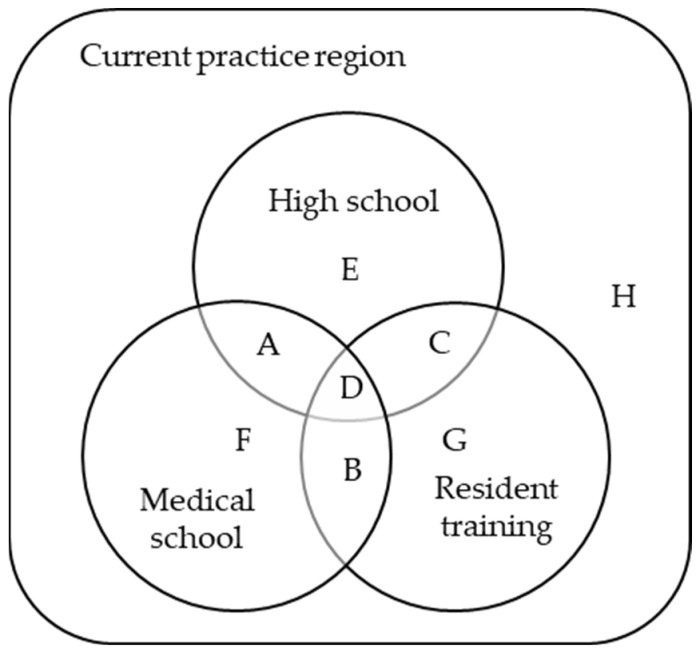
Subgroups of the study population by high school, medical school, and resident training locations. Subgroup A: those who completed high school and medical school education in the current practice region (CPR); Subgroup B: completed medical school and resident training in the CPR; Subgroup C: completed high school and resident training in the CPR; Subgroup D: completed high school, medical school, and resident training in the CPR; Subgroup E: completed only high school in the CPR; Subgroup F: completed only medical school in the CPR; Subgroup G: completed only resident training in the CPR; Subgroup H: did not complete any education course in the CPR.

**Table 1 healthcare-11-01203-t001:** Proportion of subgroups in each region.

			Current Practice Region							
			Metro	Non-Metro (6 Regions Subtotal)	Chung Cheong	Jeolla	Gangwon	Daegu/ Gyeongbuk	Busan/ Ulsan/ Gyeongnam	Jeju
Subgroup	A	*n*	15	296	66	97	8	50	73	2
		%	0.4	8.2	9.3	12.7	4.1	6.6	6.6	2.2
	B	*n*	548	175	64	23	27	26	35	0
		%	15.7	4.8	9.0	3.0	13.9	3.4	3.2	0.0
	C	*n*	409	52	6	4	1	10	31	0
		%	11.7	1.4	0.8	0.5	0.5	1.3	2.8	0.0
	D	*n*	1611	1917	213	550	23	519	612	0
		%	46.1	52.9	30.0	72.0	11.9	68.2	55.4	0.0
	E	*n*	90	182	32	29	10	26	68	17
		%	2.6	5.0	4.5	3.8	5.2	3.4	6.2	18.9
	F	*n*	8	67	33	6	14	3	11	0
		%	0.2	1.8	4.6	0.8	7.2	0.4	1.0	0.0
	G	*n*	603	87	19	1	9	8	49	1
		%	17.2	2.4	2.7	0.1	4.6	1.1	4.4	1.1
	H	*n*	213	849	278	54	102	119	226	70
		%	6.1	23.4	39.1	7.1	52.6	15.6	20.5	77.8
Total			3497	3625	711	764	194	761	1105	90

Subgroup A: those who completed high school and medical school education in the current practice region (CPR); Subgroup B: completed medical school and resident training in the CPR; Subgroup C: completed high school and resident training in the CPR; Subgroup D: completed high school, medical school, and resident training in the CPR; Subgroup E: completed only high school in the CPR; Subgroup F: completed only medical school in the CPR; Subgroup G: completed only resident training in the CPR; Subgroup H: did not complete any education course in the CPR.

**Table 2 healthcare-11-01203-t002:** Retention rates in metropolitan and non-metropolitan regions by subgroup.

		Current Practice Region													
		Metropolitan		Chungcheong		Jeolla		Gangwon		Daegu/ Gyeongbuk		Busan/ Ulsan/ Gyeongnam		Jeju	
		R	SP/S	R	SP/S	R	SP/S	R	SP/S	R	SP/S	R	SP/S	R	SP/S
Subgroup	A	46.9	15/32	40	66/165	24	97/405	25.8	8/31	26.9	50/186	36.9	73/198	50	2/4
	B	71.6	548/765	58.2	64/110	57.5	23/40	26.7	27/101	52	26/50	83.3	35/42	0	0
	C	77	409/531	66.7	6/9	57.1	4/7	100	1/1	71.4	10/14	86.1	31/36	0	0
	D	84.3	1611/1912	86.2	213/247	79.9	550/688	56.1	23/41	81.6	519/636	93.3	612/656	0	0/2
	E	39.1	90/230	17.9	32/179	9.8	29/295	20.8	10/48	14.3	26/182	20.9	68/326	27.9	17/61
	F	22.9	8/35	12.6	33/262	5.7	6/106	5.5	14/257	5.1	3/59	12.8	11/86	0	0/6
	G	58.9	603/1023	40.4	19/47	11.1	1/9	29.0	9/31	30.8	8/26	51	49/96	50	1/2

R: retention rate; SP: number of respondents who stayed in the same region as a clinician; S: number of respondents who completed high school/medical school/resident training in the region. Subgroup A: those who completed high school and medical school education in the current practice region (CPR); Subgroup B: completed medical school and resident training in the CPR; Subgroup C: completed high school and resident training in the CPR; Subgroup D: completed high school, medical school, and resident training in the CPR; Subgroup E: completed only high school in the CPR; Subgroup F: completed only medical school in the CPR; Subgroup G: completed only resident training in the CPR.

## Data Availability

The data underlying this article are available on request through the Research Institute for Healthcare Policy. https://www.rihp.re.kr/doctor/introduce.php (accessed on 1 April 2018).

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
