# Peer review of "Determinants of Working Practice Location for Clinicians According to High School, Medical School, and Resident Training Locations in Korea"

_healthcare, 2023, doi:10.3390/healthcare11091203_

Round 1

Reviewer 1 Report

I think your research is very relevant at a time when the importance of the geographic distribution of physician practices is being emphasized.

You used the 2016 KPS data, which is a representative sample of Korean physicians, and tried to prove your hypothesis using a variety of variables. Your analysis of physicians' high school graduation area, university graduation area, residency training area, and current practice area into eight subgroups is excellent. It's also great that they looked at metropolitan versus non-metropolitan areas and divided them into seven regions.

Reviewer's comments and corrections include the following.

1. You need to review and confirm Table 2 regarding the retention rates proposed by the authors. In the H subgroup, the numbers in the denominator (S) for regions other than Metropolitan are very large (e.g., 7047 for Jeju etc.).

2. Looking at Table 1, there is no footnote explanation for the subgroups (A, B, ..., H).

3. It is a minor correction, in Table 1 and Table 2, the words 'Current practice region' in the first row should be placed in the middle.

4. Minor correction.

Table S2. Comparing the number of residents to the number of medical school students in the 7 regions.

Thank you for the opportunity to review your excellent paper. 

Nothing special, partially described in the comment above.

Author Response

Dear Reviewer,

We appreciate for your precious time in reviewing our paper and providing valuable comments. It was your valuable and insightful comments that led to possible improvements in the current version. The authors have carefully considered the comments and tried our best to address every one of them. We hope the manuscript after careful revisions meet your high standards. The authors welcome further constructive comments if any. We provide the point-by-point responses, please find attached file.

Sincerely,

Hyeongsu Kim, MD PhD, [email protected]

Professor, Department of Medicine, Konkuk University

Reviewer 2 Report

Recently, the imbalance in the regional distribution of the number of doctors in Korea has become a major factor causing health inequality. Considering these characteristics, this study is considered to be helpful in determining medical manpower supply and demand policies.

However, I would like to add a description by referring to the following points in the discussion.

[The ratio of subgroup D is 49.5%, and the retention rate (SP/S) of subgroup D is high in many regions. Therefore, it seems that the key to the regional distribution imbalance of the number of doctors will be whether the SP fraction of subgroups other than subgroup D can be increased.]

A discussion of this point of view seems necessary.

Author Response

(The authors gave the same response as above.)

Reviewer 3 Report

This is an interesting report on the association / connection between various physician training location touchpoints and current work location in Korea. I like their simplified 8 categories, as per Figure 2, to deal with the multiple possible combinations. My concerns are not critical flaws, they aim to improve its translation to the audience.

A critical part of the study is allocating Korea’s locations into 7 regions, built off 17 areas. However, the authors provide no further information on these 17 areas – where do they come from & how are they defined, why are they an appropriate choice for the study, what is the justification for collapsing this from 17 to 7? In addition to these questions, why is there not a map to help visualise them? Moreover, Table S2 contains important context to these regions and the macro-level distribution of Korea’s population & physician training but is only first referenced late in the Discussion section – it should be linked into the Methods.

A minor point with language – regarding ‘retention’ the authors make statements like “remained in the region” or “continued to practice in the same region”. However, what is not known is whether the physician in fact departed a location and then returned later. This study is demonstrating associations between certain cross-sectional measures, but it cannot comment specifically on long-term retention. Also, the introduction is short with minimal references, but I also understand this is a ‘brief report’.

A weakness of the paper is its data source – the observed cohort represents <10% of the workforce. Of concern, the authors make no attempt to address or check for potential bias of their cohort – do they not have some population-level data to check their cohort’s characteristics for participant bias? Even just checking a couple of factors would strengthen the ‘trust’ that their cohort is reasonably ‘representative’ of Korea’s physician population. Confusingly, the authors mention “stratified quota sampling” but it appears that the whole cohort was sent emails requesting participation, so I can’t see where any ‘sampling’ was actually applied.

Regarding presentation of results, they have 1 metropolitan and 6 ‘rural’ regions. Combining these is problematic, I suggest removal of any aggregate results as the ‘opportunities’ for observing retention in metropolitan vs rural is vastly different. The first 2 columns of Table 1 are interesting, highlighting the substantial metro and non-metro differences. For example, subgroups A & F are almost exclusively non-metro, whereas C & G are almost only in metro regions.

In Table 2, I don’t understand subgroup H – the concept of ‘retention’ when there are no previous connections in a location seems nonsensical. Thus, why aren’t all retention rates equal to 0% in each region? This row should be removed. For all other subgroups, as per my comment above, they are observed in the same location but not necessarily measuring ‘retention’. One concern with interpreting Table 2 is that the denominator values are not previously detailed (e.g. they could be added to Table S1), thus it is somewhat confusing to read / understand at first.

I have some concerns with the author’s focus on the 2nd, 3rd & 4th highest retentions observed in subgroups B, C, and G. Whilst factually correct, these also have very small counts in the non-metro regions. In the 3rd para of the Discussion, the authors appear to be focusing on metro observed data but infer outcomes to a rural context. I think that the authors are trying to use these data as evidence to support the growth of rural-based resident training, but it is somewhat clumsy in doing so.

Author Response

(The authors gave the same response as above.)

Round 2

Reviewer 3 Report

All concerns have been satisfactorily addressed